# Revisiting the Meaning of the Divine Preservation of the Qur'an: With Special References to Verse 15:9

## Ismail Albayrak

School of Theology, Faculty of Theology and Philosophy Australia Catholic University, Melbourne 3002, Australia; ismail.albayrak@acu.edu.au

**Abstract:** In this article, I discuss the topic of Divine preservation of the Qur'an in the context of exegesis verse 15:9: 'Indeed, it is We who sent down the reminder (*dhikr*) [i.e., the Qur'an], and indeed, We will be its guardian'. First, I will briefly examine Muslim and non-Muslim perceptions regarding authentic transmission of the Qur'an. I question some Western researchers, who problematise Muslims' approaches to the history of the Qur'an with various polemics, and the Muslim perception, which compresses the notion of Divine preservation of the Qur'an into the narrow meaning of verse 15:9 with dogmatic and apologetic reflexes. Then, I chronologically analyse interpretations of the verse from linguistic, historical, theological and sociological aspects of the exegeses from the classical period to the modern times. The main argument of this article is centred on verse 15:9, which, in my opinion, deals with the first route of Qur'anic revelation from God to the Prophet through Angel Gabriel, and the Prophet's perfect memorisation of it together with his flawless transmission to the Companions. The next processes (post-Prophetic developments in transmission of the revelation) were left to the foresight of the Muslim community, *ummah*. I believe, while the vertical process (revelation from *arsh* to *farsh* (heaven to earth) is guaranteed by God, responsibility for the horizontal process (from the Companions to the following generation and so on) is left to the Muslim community. The use of the same Qur'anic text by Sunnis, Shi'ites, Ibadis, etc., from northern Kazakhstan to West Africa, from Asia Pacific to the Balkans, is the clearest proof that Muslims are preserving the book entrusted to them. They demonstrate the reflection of their understanding of *tawhid* (Oneness of God) in their faith on the Qur'an by their consensus on a single *mushaf*.

**Keywords:** the Qur'an; exegesis; Divine preservation; revelation; *mushaf*

## 1. Introduction

Recently, I came across an interesting news piece. One frightening consequence of the Australian NSW state government's long-standing land-clearing policies and plan to raise the wall of Warragamba Dam is that Australia's rare bird species (honeyeaters) will become extinct in a very short time. As a person who considers biodiversity one of the most important signs (*ayat*) of God, I want to connect this sad news with the topic of this article. According to the report, female honeyeaters reject the males' song variations because young male honeyeaters whose natural habitat is degraded cannot sing properly during the mating season. This is happening because species numbers are so low due to land clearing; with the generational gap created by this operation, honeyeaters are not properly learning the songs from their elders and are now on the brink of extinction (Vincent 2021).

Regrettably, some non-Muslim researchers treat Muslim perceptions of the history of the revelation, Qur'an and *masahif* like the plight of the poor honeyeaters. Theodor Nöldeke (d.1930), widely regarded as a leader of Qur'anic studies in the West, argues the Prophet forgot early revelations or lost his rhetorical power in Mecca when he arrived in Medina (Robinson 1996; al-Azami 2003). There are also those who struggled to find the urtext (*nuskha asliyya* or *al-umm*) for a long time because, for them, the present *mushaf* was created later (Abd al-Razzaq 2018, pp. 507–37). Besides those who sharply distinguish between

codification (*tansiq al-matn*) and canonisation (*nass qanuni*) (Khatib 2018, pp. 543–89), a small group of researchers postpone the collection of the Qur'an to two centuries after the Prophet (Wansbrough 1977). What is more interesting is that John Wansbrough (d.2002), who is considered the mentor of these researchers, determined the birthplace of the Qur'an as Mesopotamia (Wansbrough 1977; Adam 1997; Reynolds 2008). He came to this conclusion as a result of his research directly based on the Qur'anic text (literary, form, reduction, criticism, etc.), unlike the classical Orientalists, who, to some extent, use general materials from Islamic tradition in their historical-critical methodologies. Nevertheless, all other early manuscripts and inscription evidence contradict Wansbrough's controversial theory, including manuscripts found in Sana'a, the capital of Yemen. The most tragicomic approach to the history of Qur'anic revelation was put forward by writers such as late Günter Lüling (d.2014) and Christoph Luxenberg (see Böwering 2008). I think the supporters of this approach have presented no convincing evidence either in terms of language or historical data.

As for Muslims, they seem alien to their rich traditions as opposed to the formative and classical periods. This is arguably the biggest dilemma of the modern era for Muslims. Considerable literature on the history of the Qur'an and *masahif*, which has not been seen in any religious tradition, was produced by early and classical Muslim scholars. However, contemporary Muslims are not very interested in the literature in question, with the comfort of understanding that the Qur'an is the last Divine book, Prophet Muhammad is the last messenger and Islam is the last religion.[1] This perception, which considers the beginning of Islam and writing of the revelation simultaneously, does not think much about any of the processes of the *mushaf* in hand today. In this understanding, there is no difference between the current alphabets, orthography and writing style of the Qur'an and the time of the Prophet. If one looks at the books from the last two centuries and the editing activities of the genre, such as *kitab al-masahif* and *qiraat*, they will see they were primarily edited by Western scholars. Although the belief that the Qur'an is the last link of the revelatory circle does not prevent seeing the history behind it in the early and classical periods, it seems the situation in question has been reduced to pure faith by modern Muslims. Leaving aside the debates about this perception, which has changed a bit lately, attention will now be drawn to not only the faith but also the rational aspects of the issue in the context of interpretation of the 9th verse of Surah Hijr.

## 2. Analysis of 15:9—'It Is Certainly We Who Have Revealed the Reminder (*dhikr*), and It Is Certainly We Who Will Preserve It/We Will Be Its Guardian'

Before evaluating this verse (and other related verses), which constitutes the main frame of the discussion, it is useful to give a brief historical background of Surah Hijr. It was probably revealed after Surah Ibrahim during the years when the pressures on the Prophet and Muslims were intensified in the Meccan period.[2] It has 99 verses and is named after the people of Hijr mentioned in the 80th verse. The people of Hijr are the people of Thamud, the people of Prophet Salih. Most sources note all its verses were revealed in Mecca. However, some exegetes hold the view that verse 87 was revealed in the Medinan period. Surah Hijr informs us about the Qur'an being the word of God or how societies have always reacted against the revelation and prophethood in general, and how God made His prophets successful. It is like a summary of the five surahs (Yunus, Hud, Yûsuf, Ra'd and Ibrahim) that begin with the detached letters (*alif*, *lam* and *râ*) like Surah Hijr. The conclusion of Surah Hijr comprises an order that determines the ultimate goal of all these surahs (Işık 1998, vol. XVII, pp. 456–57). The main subjects of the surah are the Qur'an, revelation and prophecy, as well as the creation process of the human body and soul, the disobedience of Iblis (Satan) to God's command to prostrate, the otherworldly rewards of believers, the breadth of God's mercy, brief information about Prophets Abraham and Lot, the peoples of Ayka and Hijr, the good news given to Prophet Muhammad and the believers, and the warnings to the deniers. In addition to the historical context of the surah and verse, its textual context is also important. The relationship of verse 9 with the verses

before and after it exceeds the limits of this article, but I believe the chapter should be evaluated together, starting from verses 6 to 18. In particular, the 17th and 18th verses are directly related to the 9th verse.

Following pages, I am drawing on diverse styles of exegeses in their chronological order regarding the interpretation of this verse. Muqatil b. Sulayman (d.150/767), one of the earliest Qur'anic exegetes, makes a brief comment on verse 15:9. According to him, the meaning of the 'Divine preservation of the Qur'an' mentioned in the verse is that 'Satans cannot reach the Qur'an' (Muqatil 2002, vol. II, p. 425). Tabari (d.310/923) gives more detailed information about the verse. According to Tabari, the purpose of *dhikr* is the Qur'an in the verse, and the Qur'an has been protected from the addition of all kinds of falsehoods that do not belong to it, and the reduction in its provisions (*ahkam*) and commandments (*fard*s) (Tabari 2000, vol. XVII, p. 69). Tabari adds the nuances of different exegetical reports to his first interpretation. For example, in the narration he made from Qatada (d.117/735), he explains the falsehood (*batil*) with Satan (*Iblis*) and points out that Satan can neither add falsehood to nor diminish any truth in the Qur'an (Tabari 2000, vol. XVII, p. 69). With this interpretation, Tabari makes an inter-textual reading by referring to another verse in the Qur'an: 'No falsehood can approach it from before or behind it: It is sent down by One full of wisdom, worthy of all praise' (41:42). This explanation by Tabari has become the norm for later exegetes and many commentators have repeated this interpretation word for word. Tabari also mentions the alternative meaning with the formula of *qila*—'it is said'. Accordingly, the meaning of the verse is 'Indeed, it is We who sent down the reminder [i.e., dhikr, the Qur'ān], and indeed, We will protect Muhammad from his enemies' harm' (Tabari 2000, vol. XVII, p. 69).

Zajjaj (d.311/923), Samarqandi (d.373/983), Tha'labi (d.427/1035), Mawardi (d.450/1072), Qushayri (d.465/1072), Wahidi (d.468/1076) and Baghawi (d.516/1122) mention similar interpretations (Zajjaj 1988, vol. III, p. 174; Samarqandi n.d., vol. II, p. 25; Tha'labi 2002, vol. III, p. 331; Mawardi n.d., vol. III, p. 149; Wahidi 1994, vol. III, p. 40; Baghawi 2000, vol. III, p. 51; Qushayri n.d., vol. II, p. 264). Nevertheless, there are some nuances among their presentations. For instance, Mawardi divides *hifz* (protection) into three categories and explains the last one as 'God keeps the Qur'an in the heart of those who wish good, and not those who wish evil, *sharr*' (Mawardi n.d., vol. III, p. 149). While Tha'labi says they do not change letters from the Qur'an (Tha'labi 2002, vol. III, p. 331), Qushayri points to the 44th verse[3] of Surah Maidah and states God directly authorised the Israelites to protect their scripture, but they abused this authority and falsified and altered the Torah. As for the Qur'an, Qushayri argues that God has advised He will protect it forever. He points to the heart of the *qurra*s (who memorised the Qur'an) as the treasure of His book and says nothing will be lost from the Qur'an (Qushayri n.d., vol. II, p. 264). Baghawi also mentions verse 5:67[4] about the alternative view, reiterating Tabari's comments. Traditionalist Ibn Kathir (d.774/1373), on the other hand, indirectly states the meaning derived from the textual context of the verse is the Divine preservation of the Qur'an (Ibn Kathir 1998, vol. IV, p. 527); therefore, he does not find the alternative meaning (protection of Prophet Muhammad from his enemies) regarding exegesis of this verse.

It is useful to look at couple of classical Shi'ite commentators' approaches to the verse 15:9 The Shiite scholar Abu Ja'far al-Tusi (b. 460/1067), known as *Sheikh al-Taifa* (the leader of the group, Sheikh), does not interpret the issue differently from his Sunni counterparts. Tusi, who mentions the accepted and alternative meaning together, states that the Qur'an is preserved from additions, removal, and deficiencies. However, it is seen that Tusi briefly draws attention to an interesting dogmatic issue in this verse. According to him, this verse is proof that the Qur'an was created later (*huduth al-qur'an*). He argues that if something was revealed (*munzal*) and preserved (*mahfuz*), it was created later (*muhdath*). Because he thinks that protection is not permissible for the Eternal (*qadim*) and it will not be needed. Like his Sunni exegetes, Tusi also reads 15:9 in the light of other and similar verses in the Qur'an (Tusi 2022, *Tibyan*, Online access August 2022). Famous exegete Tabarsi (d.548/1154), who was born ten years after Tusi's death, closes the issue by repeating more or less what Tusi

had already said and that the Qur'an was created later (*muhdath*) ([Tabarsi 2006](), vol. VI, p. 80).

So far, it can be seen that commentators have not explained the whole context of the event of the revelation within the framework of their general comments. It is as if, in their minds, there is acceptance that God preserves the text of the Qur'an with direct intervention, even though the nature of this protection is not much elaborated. Sometimes it is brought to the agenda that God protects His word through His servants who have memorised the Qur'an. However, they do not provide any explanation that collectively deals with the verses at the beginning of Surah Hijr and draws attention to some nuances. As will be discussed later in more detail, I argue the problem of those who question the Prophet in the textual context of this verse (15:9) is related to the route of the revelation from God (heavenly realm) to the Prophet (earthly realm). Otherwise, it is seen that people who questioned the Prophet about the origin and authenticity of the revelation have no problem that the Prophet conveyed the revelation he received from God correctly and those who learned it from the Prophet would memorise, preserve then pass it onto the next generations. In other words, when a two-stage transmission of revelation is considered, the first stage is mentioned in this surah. From these stages, I do not mean the well-known interpretation of Ibn Abbas (d.68/687–88) who considers the process first from *lawh al-mahfuz* (heavenly preserved tablet) to the closest heaven (*al-sama al-dunya*) in total, then from the closest heaven to the Prophet in piecemeal ([Tuncer 2021](), pp. 423–34; [Nasa'i 1992](), vol. I, p. 69). The first step of the two-stage transmission of the revelation is to be taken from God and delivered to the Prophet intact, and the Prophet conveyed the revelation he received while he was alive to his Companions. The second stage is about the transmission of the Qur'an to the generations after the Prophet by the Companions and followers and so on.

In fact, we see a similar approach not only in commentators but also in different Islamic disciplines dealing with this subject. For example, the Divine protection of the Qur'an entered the *raddiya/milal* and *nihal* (polemic and rejection) literature of the early periods in the form of the superiority of the Qur'an and therefore of Islam. These scholars argue that God assumes the role of protector of the Qur'an with different expressions in light of verse 15:9. Some use the expression *li-annahu ta'ala takaffala bi-hifzihi*, while others prefer to say *fa-tawalla Allahu hifzahu, tawalla hifza kitabihi bi-nafsihi al-karimati, wa-lam yakil dhalika ila ghayrihi* (God took over the protection of the Qur'an and did not delegate this task to anyone) ([Sijistani 1998](), vol. I, p. 1; [al-Hawali n.d.](), vol. I, p. 720; [Ibn Battal 2003](), vol. VIII, p. 74). Ibn Qutayba (d.276/889), on the other hand, gives an interpretation to the fact by stating God gives the responsibility of preserving the Qur'an to those who will protect the book of God from doubt and transmit it sensitively with uninterrupted chains ([Ibn Qutayba 1999](), vol. I, p. 9). Isfarayani (d.406/1016) carries the subject completely to the dogmatic field and argues the protection mentioned in the verse is actually the protection of the Qur'an from internal contradiction ([Isfarayini 1983](), vol. I, p. 186). Thus, in the periods when formative and early classical works were produced, there was no clear-cut distinction between vertical (top down) and horizontal (parallel) transmission of the revelation, or we observe there is no problem in minds in this regard.

In later periods, we witness that both stages are mentioned together more frequently in exegetical works. The first two examples on this subject are two contemporary commentators from the classical period: Mutazilite theologian and exegete Zamakhshari (d.538/1144) and Andalusian exegete Ibn Atiyya (d.541/1147). They make comments in a way that summarises all previous exegeses. Rationalist Zamakhshari connects verse 15:9 with the sixth verse of the same surah and draws attention to the fact God sent Gabriel to the Prophet with the Qur'an, and while He was sending it, watchers and guards (*rasad*) were around Gabriel—in front of and behind him. He states the guardian protecting the Qur'an from all kinds of distortions, alterations and omissions is always God, since the watchers with the permission of God protect the revelation from Satan and ensure it safely reaches the Prophet. For this reason, according to Zamakhshari, there is no deficiency or excess in

the Qur'an. He also polemicises about the transmission of other Divine books by saying God did not guarantee their preservation. On the contrary, for Zamakhshari, God gave this task to the rabbis and priests (*ahbar* and *rabbaniyyun*) but they falsified their respective scriptures. With this explanation, Zamahshari alludes to verse 5:44, whose translation is given above. Zamakhshari does not neglect to give the alternative meaning of verse 15:9 with the formula of *qila* (it is said), the preservation is the protection of the Prophet (Zamakhshari 1987, vol. II, p. 572).

Ibn Atiyya explains the verse starting with the alternative meaning. He mentions the main meaning, which he calls the opinion of the majority, as the second opinion. Drawing attention to similar issues with Zamakhshari, Ibn Atiyya makes a statement regarding the interpretation of the verse: 'According to the narration from Ibn Abbas on the last page of Bukhari's *al-Sahih*, it is said that the distortion in the previous books is not textual, but interpretive, whereas (according to Ibn Atiyya) the explicit expression of the verses speaks of textual distortion' (Ibn Atiyya 2002, vol. III, p. 351). In this period, when it became the norm to explain the preservation of the Qur'an by falsifying previous books, Ibn Atiyya, like many commentators, refers to verse 41:42 and states the Qur'an is preserved in the chest (heart) of the people (lit. men) (Ibid). It is safe to assume Ibn Atiyya's interpretation resembles the sensitivity of Andalusian commentators towards the *isra'iliyyat* reports. In other words, unlike his Eastern colleagues, the polemical element predominates Ibn Atiyya's comments, since he lived in Andalusia where there were many non-Muslims. It is as if Ibn Atiyya was more concerned with the falsification of other religious texts than with proving the preservation of the Qur'an in the interpretation of verse 15:9.

The versatile exegete Razi (d.606/1210) deals with verse 15:9 in the most comprehensive and systematic way. He first draws attention to the pronoun 'We' in the verse 'We who sent down the message [i.e., the Qur'ān], and indeed, We will be its guardian'. Razi rationalises God's use of the plural pronoun and likens such expressions to a king saying, 'we did it' instead of 'I did it'. Thus, a preferred rhetoric usage (plural) indicates the greatness and magnificence of the speaker, namely God. The second subject Razi focuses on is the determination of the pronoun 'its' in the expression 'We will be its guardian'. As it is often mentioned above, the first view is to associate this pronoun with the previously mentioned word *dhikr* (message, reminder: the Qur'an). The meaning is to say there is neither a deficiency nor excess in the Qur'an, and it has been protected from every kind of distortion. At this juncture, Razi reminds the reader of two verses—4:82[5] and 41:42 (Razi 1999, vol. XIX, p. 123). Razi raises a hypothetical question with the same formula (*fa-in qila*, if it is asked or said): If God had guaranteed the preservation of the Qur'an, why would the Companions have bothered to collect the Qur'an in the *Mushaf*? They should not have been afraid (that the Qur'an would be lost). Razi answers his question with: *anna jam'ahum li al-qurani kana min asbabi hifz Allah ta'ala . . .* /'their collection of the Qur'an is one of God's means of preserving it, and Allah has given the responsibility to the Companions in this regard' (God prompted or motivated them to collect the Qur'an) (Razi 1999, vol. XIX, p. 123).

Razi then discusses another topic, although this subject, which he has mentioned as a parenthetic clause, is not directly related to the issue being discussed in this article. It is more related to the intra-juristic schools' competition. While Razi is trying to support his juristic school, he also opens the issue a little further in the exegesis of verse 15:9. In other words, he is trying to hit two birds with one stone. Razi argues this verse constitutes strong evidence for the Shafiite's view that the *basmala* at the beginning of each surah is a part of the Qur'an. The meaning of the Qur'an being preserved by God is that it remains protected from excesses and deficiencies. Therefore, if the *basmala* was not from the Qur'an, the Qur'an would not have been protected from changes and excesses. (Razi 1999, vol. XIX, p. 123). Razi, perhaps unwittingly, accuses his coreligionists and some Qur'an reciters (*qurra*) who do not consider *basmala* a full verse in each surah other than at the beginning of Surah Fatiha, but his linking the issue with verse 15:9 provides important clues for an understanding mindset.

Under the second issue, Razi evaluates the alternative meaning of the verse (protection of the Prophet) as allegorical usage. According to him, *inzal* (descending) and *munzal* (descended) were mentioned, and he finds it appropriate to mention *munzal alayh* (descended upon) in the verse. However, he goes on to state the textual context supports the first interpretation and this first one is preferred and better: *anna al-qawla al-awwal arjah al-qawlayna wa ahsanuhuma* (Razi 1999, vol. XIX, pp. 123–24).

Razi's third issue seems to be the commentary of the second explanation. Razi, who starts by bringing up the controversy about how God protects the Qur'an, tries to persuade the addressee by using dogmatic and rational arguments. According to Razi, who made the first comment by saying 'some said,' without giving a name, any additions or omissions made to the Qur'an can be direct and adversely affect its structure, *nazm*. Thus, every intelligent person understands the additions and subtractions made afterwards cannot be from the miraculous Qur'an. He likens it to the walls surrounding and protecting a city (Razi 1999, vol. XIX, p. 124). Razi implies, if the walls have cracks or holes, protection of the city will not be possible; similarly, any addition or deficiency made to the Qur'an will damage its integrity. Razi's reply to the above-mentioned question again comes from anonymous people: 'God has protected His book from the capacity of opposition (*mu'aradah*) of His creatures' (Razi 1999, vol. XIX, p. 124). Another group says the Qur'an has rendered human beings incapable of annulment or corruption and has sent a community (group) to protect its book by constantly studying and teaching others until the Day of Judgment. He points out that even an experienced sheikh would be warned by a little child when he misread the Qur'an. For Razi, no book has this kind of privilege and preservation. It probably includes the theory of Mu'tazila *sarfa* (God destroys people's attempts to imitate the Qur'an or He destroys their will to do so). Another of Razi's comments presents information on the preservation of the Qur'an in the context of Qur'anic *i'jaz*, inimitability of the Qur'an that includes inter-religious polemics and puts the issue on a rational basis: Many Jewish and Christian deniers (*mulhid*s) have lauded their efforts in the name of the annulment and corruption of this greatest miracle of God. Despite the fact it has been preserved for about 600 years, it is a foretelling of the unseen (*ghayb*) and an obvious miracle. Razi's last discussion is based on Shiite–Sunni polemic in the light of Mu'tazilite Qadi Abd al-Jabbar's (d.415/1025) argument. Razi begins with a statement by saying al-Qadi made such an inference, then criticises Imamiyya Shi'ites and Qadi Abd al-Jabbar for the inference, by bringing up the claim that some people from the Shi'ite Imamiyya made additions or subtractions to the Qur'an (Razi 1999, vol. XIX, pp. 123–24). Razi appears to be one of the first commentators to deal with all aspects of the subject and in a more rationalistic way.

### 3. The Understanding of the Verse during Post-Razi Period

We are witnessing various approaches in this period as well. For instance, Qurtubi (d.671/1273) prefers a different path in his interpretation of verse 15:9. In his introduction, he briefly states God has taken the preservation of the Qur'an on Himself. However, God left protection of the other Divine books to the clergy; therefore, these scriptures were exposed to distortion and alteration. Then he narrates a long anecdote (event and mentioning many narrators' names—men and women) that took place during the reign of Caliph Ma'mun (d.218/833): A handsome and fragrant Jew came to Ma'mun's knowledgeable people's assembly wearing a beautiful outfit. He gave a wonderful speech, and when the conversation was over in the assembly, Ma'mun called him by saying 'O! The Jew/Israili'. The Jewish man said 'yes'. Ma'mun asked him to become a Muslim and promised to give him extra favour. The Jewish man said 'my religion is the religion of my forefathers' (expressing his dissatisfaction with converting to Islam) then left. One year later, this man again joined in Ma'mun's assembly but as a Muslim. He gave a very good speech on jurisprudence. Ma'mun was surprised, called him and asked if he was the person who did not accept the Caliph's request a year ago. When the Jewish man said 'yes', Ma'mun again asked him what happened. The Jew, after leaving the assembly of Ma'mun the previous

year, reproduced three copies of the Torah and made some minor additions and deletions, then went to the synagogue (Qurtubi says *kanisa*/church in his exegesis), where people bought these copies without noticing the additions and deletions. Likewise, the Jewish man produced three copies of the Gospel and sold them to Christians. When he went to the church, he saw the Christians did not notice the changes in their scriptures. Finally, he produced three copies of the Qur'an and took them to second-hand booksellers.[6] He says, when they flipped through the pages, they found the mistakes, did not buy them and threw the *mushaf*s on the ground. Then the Jewish man said that he decided the Qur'an was preserved and consequently he became a Muslim (Qurtubi 1964, vol. X, pp. 5–6). Qurtubi does not stop here. He goes on and says Yahya b. Aktham went on a pilgrimage that year and saw the great scholar Sufyan b. Uyayna. He told him about the event in Ma'mun's assembly and Sufyan said 'this is how it is explained in the word of God' and recited verse 41:42. Qurtubi concludes his explanations by pointing out the alternative meaning and some linguistic issues (Qurtubi 1964, vol. X, p. 6).

Unlike Razi, Qurtubi makes all his arguments through the detailed story he has chosen, rather than directly engaging in theological discussions. He also does not base his argument on Qur'anic verses. The social experience of the Jewish man, according to Qurtubi, is the clearest and strongest evidence of preservation of the Qur'an. Baydawi (d.685/1286), who is considered the crown jewel of the *madrasah tafsir* curriculum, concisely says the Qur'an is different from human speech and a linguist (*ahl al-lisan*) will notice the slightest change in the verses (Baydawi 1987, vol. III, p. 207). According to Baydawi, the Qur'an is a miracle of language, and no one can distort it. Unlike Qurtubi, Baydawi directly clings to the Qur'anic text and language (internal evidence) in his argument. The exegete Khazin (d.741/1341), who lived a generation after Baydawi, brings up Razi's comments with his own words. According to Khazin, not only humans, but even jinn cannot make any additions or deletions to the Qur'an. He reminds us of all about the alternatives, from the theory of *sarfa*, which Razi also mentions, to the fact God prompts knowledgeable scholars (*rasikhun*) to protect the Qur'an (Khazin 1984, vol. III, p. 49).[7]

Abu Hayyan (d.745/1344) first presents the notion of Divine preservation as protection from the interventions of Satan, then states God took on protection of the Qur'an and left the other Divine texts to the clergy; therefore, they were distorted. Abu Hayyan, however, identifies preservation of the Qur'an with preservation of Islamic *shari'ah* or the universality of the religion of Islam. He narrates from Hasan (probably Hasan Basri, d.119/728): 'God preserved the Qur'an by making *shari'ah* permanent until the Day of Judgement' (Abu Hayyan 1999, vol. VI, p. 468). Abu Hayyan does not neglect to mention the alternative meaning (Abu Hayyan 1999, vol. VI, p. 468).

Hanbali scholar Ibn Rajab (d.795/1393) dealt with the subject in detail during the post-Razi period. Ibn Rajab argues, when a prophet's term expires, God renews His religion with a new prophet and this practice continues until the last prophet. According to Ibn Rajab, God takes on the protection of the last religion. At this juncture, he draws attention to the famous prophetic report that in every century there will be people who will shoulder this religion and protect it from the excesses of falsifiers, plagiarists, and ignorant interpretations. After expressing the protection of religion, he goes on to say that God took over the protection of His book just like His last religion. In this way, God states that He protects it from any addition or deficiency. Contrary to many commentators, it is observed that Ibn Rajab conveys the details mentioned in the sources about the preservation of the Qur'an. For example, he states the Prophet had his community read the Qur'an in different letters (*ahruf*) for convenience. He reports, after the spread of Islam, when Muslims came together in different places, there was conflict between them due to the prophetic permission (*ahruf al-sab'a*) of seven letters in the recitation of the Qur'an. Ibn Rajab states the Prophet's Companions agreed on one letter during the caliphate of Uthman and other *mushaf*s were burned, because they were afraid of conflicts in their own books like the disagreements of other religious communities. Ibn Rajab presents this work as one of Uthman's greatest favours (*min mahasin Uthman*). Ibn Rajab, who states *maslahat* (common

good) requires it, does not neglect to mention that Ali (d.40/661), Huzayfa (d.36/656) and many Companions supported Uthman's (d.35/656) initiative (Ibn Rajab 2001, vol. I, p. 603).

Ibn Rajab also mentions some recitation conflicts between the Companions, such as between Umar (d.23/644) and Hisham (d.15/636), Ubayy (d.33/654) and others. Then, he asks what would have happened after the Prophet? According to Ibn Rajab, the necessity of collecting (reducing the seven-letter licence to one letter) during the Uthman's caliphate is needed not only from a religious viewpoint, but also from a rational viewpoint. Therefore, for him, the *Mushaf* of Uthman was the only measure. When this is the case, if a person prefers the reading style of Ibn Mas'ud (d.32/652–3) even though he knows the consensus of the community is on the Uthman *Mushaf* or if one claims that Ibn Mas'ud recitation style (*harf*) is better than Zayd b Thabit's (d.45/665) *harf*, in the eyes of Ibn Rajab, this person is an oppressor, has crossed the line and deserves punishment (Ibn Rajab 2001, vol. I, pp. 603–4).

As it is seen, Ibn Rajab makes his argument based on this case and argues the consensus of the Muslim community on Uthman's *Mushaf* is the main measure of protection of the Qur'anic text. According to him, the preservation of the Qur'an is directly related to the measures and efforts taken by the community in the very early period. Although it is controversial that seven letters (*ahruf al-sab'a*) were reduced to one letter in the Uthman *Mushaf*, it gives a better understanding of the process and Divine protection by mentioning the details about the history of the Qur'anic text. It is possible to see a similar approach in the exegesis of Biqa'i (d.885/1480). Although, in the introduction to the exegesis of verse 15:9, he states God Almighty will preserve the Qur'an with His knowledge and power despite the envious people. He does not stop here and gives various details about the collection of the Qur'an for the first time during the reign of Abu Bakr. Biqa'i includes the following statements by Umar, who told Abu Bakr about the martyred Qur'an memorisers (*qurra*) in the battle of Yamama: 'I'm afraid that many reciters will be martyred in the ongoing wars and a large portion of the Qur'an will be gone (forgotten), but if we put it between two covers. the situation will be different' (Biqa'i n.d., vol. XI, p. 22). Biqa'i points out that Abu Bakr, who is mentioned in the report, first reacted to Umar's suggestion, was persuaded (God expanded his heart), then He convinced Zayd b. Thabit to carry out this mission. Biqa'i also describes the Divine protection without minimising the community's efforts.

While summarising the general approaches to verse 15:9, two Ottoman commentators (middle and late Ottoman period), Abu al-Suud (d.982/1574) and Shihab al-Din Alusi (d.1270/1854), draw attention to the nuances. Abu al-Suud discusses that the verse rejects the claims of those who mock the Prophet, accuse him of insanity and deny he received revelations (Abu al-Suud n.d., vol. V, p. 68). This is a really important point because, according to Abu al-Suud, the mere addition or omission of the Qur'an in the previous interpretations is not appropriate for the textual context and situation (*muqtadayi al-maqam*). According to Abu al-Suud, the Qur'an has been protected from all kinds of slander and attack. He does not completely ignore the Qur'anic *i'jaz* or other alternatives, but with this interpretation he shows that he takes the issue from a much broader perspective. Finally, Abu al-Suud, who interprets *la-hafizun* as a noun phrase with continuity, also emphasises the continuity of preservation of the Qur'anic text in this way (Abu al-Suud n.d., vol. V, pp. 68–69).

Alusi begins his comment with a short Persian quote from Husain al-Kashifi (d.910/1505): ' . . . this book brings honour to the readers'. Then he repeats Abu al-Suud's general approaches to the verse (Alusi 1994, vol. IV, p. 443). Alusi's explanations of the verse are not limited to the above couplets. In his explanations in Persian and Arabic, he argues the Qur'an is protected from the delusions and confusions of Satan. The book *Bahr al-Ulum*[8] records that the preservation of the Qur'an is explained in the form of the '*sarfa* theory' and goes on to say: 'People had the power to distort and reduce the Qur'an, just like the Torah and the Gospel texts, but God prevented them from doing this (*sarafahum*) by the scholars' compilations of books explaining the words and meanings of the Qur'anic text' (Alusi 1994,

vol. IV, p. 444). Here, Alusi means the books of exegesis, *qiraat* (variant readings), etc., were written for the sake of the Qur'an's preservation. After pointing out this, he takes a long part from Mawlana's (d.672/1273) *Mathnawi*:

> God's grace promised Mustafa (Prophet) [as]; Even if you die, this book will never die (disappear); I honour your greatest miracle (the Qur'an); I will prevent anyone who tries to reduce something from the Qur'an and add something to it . . . You don't seek a better protector than me. (Alusi 1994, vol. IV, p. 444)

Alusi also mentions the famous *mujaddid* (renewalist) hadith that will come in every century and draws attention to the existence of skilful scholars, *qurras* (reciters) and *huffaz* (those who memorised the Qur'an). Although Alusi refers to the rumour that the Qur'an will disappear from lines of the books (*sutur*) and chests (*sudur*) of people in the End Times, his general approach is similar to what Razi said before. That is, God will protect His book through His righteous servants (Alusi 1994, vol. IV, p. 444).

## 4. Evaluation of Divine Protection of the Qur'an in Modern Period

When we come to the modern period, it is observed that most exegetes repeat the classical interpretations on the subject. However, there are a few exceptions. In this context, three important names will be emphasized. The approach of Sayyid Qutub, Tabatabaei and finally Said Nursi will be analyzed.

Sayyid Qutub (d.1966) deals with the issue from socio-political and theological perspectives. He argues that verse 15:9 is a Divine miracle about the preservation of the Qur'an. Otherwise, according to Qutub, it is not possible for the Qur'an to reach us without being corrupted. Simply, for Qutub, if we do not accept an external power other than human will, the Qur'an cannot be protected safely after all these times, turmoil, countless mischief and events in the history of Islam. Qutub's general argument, on the one hand, draws attention to Divine will in preserving the Qur'an, and on the other hand, presents scholars' efforts (Qutub 1991, vol. IV, p. 2127). After that, Qutub historically touches on the events of corruption in the period from the first years of Islam to the present and says that nothing happened to the Qur'an even though there were countless difficult times. Qutub first mentions the Jewish factor, which he considers the head of every kind of evil. He also argues that tribalism, called *shu'ubiyya* (classic version of nationalism), came among the Muslims, with each group making fabricating hadiths for their own benefit and interpreting the Qur'an in a way that would suit their interests. Nonetheless, he says all these groups' efforts are in vain because the *ulama* developed the *isnad* system sensitively and they cleaned up these excesses that were later included in the hadiths. When the Qur'an is the subject, Qutub is certain that enemies of Islam were and are incapable of adding something to the Qur'an or falsifying it.

Despite this, Qutub interprets verse 15:9 as Godly evidence (*hujjah rabbaniyya*) regarding the preservation of the Qur'an. He carries his assessments to the contemporary period and frames the situation of the Islamic world and Muslims in modern discourse:

> A new time has come upon Muslims that they have been still suffering from the troubles of this new era. In this new period, their power to protect themselves, their beliefs, state systems, lands, honour, property, even their mental health and understanding has completely weakened. The victorious enemies changed all the good deeds that Muslims had and replaced those good deeds with *munkarat* (bad deeds). They placed all evil in belief, thought, virtue and value, custom and morality, law and system. By corrupting all human characteristics, they caused degeneration and immorality among Muslims. (Qutub 1991, vol. IV, p. 2127)

Qutub raises his tone and says the new life of the Muslims resembles that of animals and even animals have turned to a way of life they detest when they look at Muslims. Qutub also records that the enemies of Islam presented all these immoral and evil deeds under respectable titles such as progress, development, secularism, science, freedom, breaking the chains, revolution, and innovation. Qutub argues that nothing is left from

the religion of Islam in the hands of Muslims. He states Muslims have been swept away like foam and they have neither the power to defend themselves nor to make changes or revival. Despite all these weaknesses and desperation of Muslims, he thinks the enemies of religion cannot distort the Qur'an even though they want it so desperately and eagerly (Qutub 1991, vol. IV, pp. 2127–128); therefore, the verse miraculously proves its truth. In other words, according to Qutub, the preservation of the Qur'an, which stands firmly in the hands of Muslims who could not protect their material and spiritual possessions in the face of colonial forces and the tricks of the Jews, cannot be explained other than through Divine protection. Qutub's somewhat polemical and ideological explanations about the verse seem to evoke the preservation of the Qur'an through direct Divine intervention.

The contemporary Shiite scholar Tabatabaei (d.1981) is perhaps the most comprehensive commentator of the verse 15:9 in Muslim exegetical tradition. He explained the verse under the title of *kalam fi anna al-qur'an masûn an al-tahrif fi fusulin* (sections on the protection of the Qur'an from distortion). He allocates seven subsections, which totals 32 pages, for the explanation of this verse (Tabatabaei n.d., vol. XII, pp. 101–33). It requires a separate article to evaluate the extensive analysis of Tabatabaei's exegesis. Here we will only try to summarize his general approach to the subject. He indicates clearly in his introduction to the interpretation of the verse that the Qur'an is *dhikr* (remembrance) of God therefore it is protected against distortions in all its parts and in every respect, *fa al-âyatu tadullu ala kawni kitab Allahi mahfuzan min al-tahrifi bi jamii aqsamihi min jihati kawnihi dhikr Allah subhanahu*. For Tabatabi, the Qur'an is a living and immortal *dhikr* that is always alive and, on the agenda, (Tabatabaei n.d., vol. XII, p. 101).

Tabatabaei deals with the subject from historical, literary, socio-cultural, theological and methodical perspectives in each section and gives special importance to naming of the Qur'an as *dhikr* in the verse. According to Tabatabaei, the concept of *dhikr* is the most comprehensive attribute that points out the effect and function of the Qur'an. For this reason, it is mentioned as *dhikr* in the verses that describe that the Qur'an is protected against attempts to falsify. The verses 40–42 of Surah Fussilat is a good example in this regard. In 15:9, God both used the word *dhikr* in absolute sense and also, He stated the meaning of the protection is an absolute. It is understood from this that, with the absolute preservation of God, the Qur'an is protected against all kinds of additions, reductions, and textual alterations in such a way that it will remove the feature of being *dhikr* and will somehow end it from being the dhikr of God (Tabatabaei n.d., vol. XII, p. 107).

In summary, Tabatabaei says that in principle, the Mushaf we have is the same Qur'an that was revealed to the Prophet. According to him, if something is missing from the Qur'an revealed to the Prophet; If it were assumed that there was a change in its wording, letters or sentence structure, it would have to be in situations where the Qur'an would not be able to function in any of its qualities such as being a miracle, eliminating contradictions, leading to the right path, being a light (*nur*) and *dhikr*, and dominating (*al-haymana*) the previous heavenly books. However, Tabatabaei does not completely rule out that some insignificant or minor changes may occur. For example, dropping the point of a letter or a repeated verse, or mini details in the grammatical structure (diacritical etc) of the word. (Tabatabaei n.d., vol. XII, p. 107).

Tabatabaei argues that the excellent literary (rhetorical, eloquence and *tahaddi* verses) structure of the Qur'an, the authentic information it gives about the previous prophets and their people, the existence of sound law (*shari'ah*) and provisions (*ahkam*), and the absence of contradictions are the important evidence for the preservation of the Qur'an. In addition, he discusses that the existence of a famous *thaqalayn* hadith in Sunni and Shiite sources, the Imams of Ahl al-Bayt unceasing efforts in ensuring the authenticity of every hadith by comparing them with the Qur'an, and the existence of numerous reports about the frequent recitation of the Qur'an by the Prophet are also important proofs for the preservation of the Qur'anic text. Tabatabaei argues that neither the opposition nor the supporters bring up the possibility of falsification of the Qur'an. For him this is clear evidence for the protection of the Qur'an (Tabatabaei n.d., vol. XII, p. 124).

It is also important to note that Tabatabaei discussed various contradictory narrations and problematic isnads in the Shiite and Sunni sources on the subject in a very rational basis. For example, according to Tabatabaei, many narrations quoted in Shiite and Sunni sources state that some suras and verses, as well as sentences or parts of sentences, words and letters in these verses were dropped and changed during the collection process (Tabatabaei n.d., vol. XII, pp. 109, 113). It is worth dwelling on his analysis of how it should be evaluated. Tabatabaei's view on the subject is not straightforward. He argues that the reports are generally single-channel (*âhâd*) narrations supported by certain presumptions and definitively (*qat'i*) reject any distortion in the sense of adding and changing the Qur'an. As for distortion in the sense of reduction, Tabatabaei says that the reports negate it as an uncertain (*zannî*) way (Tabatabaei n.d., vol. XII, p. 126).

Tabatabaei also says that the verses, passages or chapters, which are stated to be a part of the Qur'an in many narrations, contain contradictions within themselves and in no way resemble the style, manner, genre and content of the Qur'anic text. He cited Sunnis' *qunut* prayers *(khal'* and *hafd*) and Shiites' famous Surah of Walayat (friendship) as examples in this regard (Tabatabaei n.d., vol. XII, p. 115). Furthermore, Tabatabaei's evaluations about the stoning verse (*rajm*) and the lengths of some surahs in some reports are very important. According to him, these reports point out that the recitation (*tilawat*) of stoning verse is abrogated and the narrations that a certain surah used to be much longer in the past (Surah Ahzab is longer than Surah al-Baqara and two hundred verses of this surah have been lost, etc.) are completely contradictory to the Qur'an (Tabatabaei n.d., vol. XII, pp. 116–67).

Another point that Tabatabaei makes is the reports that refer to the extra statements and additions pointing to the Ahl al-Bayt Imams in the Qur'anic text, especially in the Shiite sources. For example, the expressions which runs as *ya ayyuha al-rasul balligh ma unzila ilayka fi Aliyyin* mentioned in the the verse 5:67 is a good example in this regard. Tabatabaei evaluates this addition (*fi Aliyyin*) in the verse and many similar reports as drawing attention to the occasion of revelations (*asbab al-nuzul*) rather than authentic additions to the verse of the Qur'an. In other words, he considers this addition as the explanation of the verse about Imam Ali rather than seeing it as a part of the Qur'an (Tabatabaei n.d., vol. XII, p. 113). In another place, there are reports from Imams that imply that the Qur'an has been corrupted. For instance, if someone said, 'This verse was actually revealed in this way', this does not mean that the verse wording is exactly what Imam is referring to. For Tabatabaei, such words of these Imams are basically a type of interpretation of the verses, and this was said in response to the esoteric (*batini*) and deep hermeneutics (*ta'wil*) of the verse (Tabatabaei n.d., vol. XII, p. 108).

Another issue Tabatabaei has mentioned is the Mushaf attributed to Imam Ali. While examining this issue, which he has brought up several times, the subject he focuses on is the difference between the Mushaf of Ali and other Masahif. He only accepts the difference between two as the arrangement of some special verses and suras. In short, these differences have nothing to do with the fundamental truths of religion (Tabatabaei n.d., vol. XII, pp. 108, 117). In fact, he states openly that the different Qur'anic suras and the order in some verses gathered together between the two covers cannot be completely isolated from the *ijtihad* (personal opinions and practices) of the Companions. Tabatabaei argues that there is no direct evidence from the Prophet regarding the arrangement of the verses and suras of the Qur'an (Tabatabaei n.d., vol. XII, p. 124). For this reason, he does not accept the notion of pre-existence of the Qur'an in the *lawh-i mahfuz* (protected tablets) and also does not believe that it is arranged in accordance with the format of the Qur'an which is found in the closer heavens (*sama al-dunya*) of the world (Tabatabaei n.d., vol. XII, p. 131). Although he does not deal with the notion of createdness of the Qur'an (*muhdath*) like his predecessors Tusi and Tabarsi, it is safe to assume that with this explanation he confirms this understanding.

Tabatabaei in his evaluations of the verse 15:9 presents many interesting perspectives, poses questions and offers answers to these questions. One of the most interesting questions is as follows "is it possible for ordinary people to put together scattered and fragmented

pieces of the Qur'anic text and finalise the process of collecting the Qur'an without error?" (Tabatabaei n.d., vol. XII, p. 116) Here he is questioning about the possibility of the collection of the Qur'an by normal people completely and in accordance with the truth without the involvement of *ma'sum* Imams (free from sin or innocent Imams). Tabatabaei suffices to say, in short, that reason (and wisdom) judges that this is possible (Tabatabaei n.d., vol. XII, p. 116). He also problematises the reports about the previous religious communities in the light of some narrations. According to these reports Muslim societies will be tested similar to the Jews and Christians who went through before. As the report notes 'this ummah (Muslim) will follow step by step what the Children of Israel went through' and that Muslims will distort the Qur'an just as they distort their holy books (Tabatabaei n.d., vol. XII, pp. 111–12). Tabatabaei is very clear on this issue: "he Qur'an has not experienced similar calamities that befell the Torah, the Gospel and other heavenly scriptures that were revealed to other prophets" (Tabatabaei n.d., vol. XII, p. 125). Tabatabaei's most comprehensive assessment on the subject is as follows: 'It is understood from these detailed explanations that the Qur'an, which God revealed to His Prophet and described as preserved as it was revealed, is protected against all kinds of additions, deficiencies and changes with the protection of God. As a matter of fact, Almighty God promised His Prophet in this regard and gave this guarantee to him' (Tabatabaei n.d., vol. XII, p. 107).

Finally, it is worth remembering that Tabatabaei considers the alternative meaning of the verse (the pronoun la-hu/him is directed towards the Prophet' as highly inaccurate (Tabatabaei n.d., vol. XII, p. 106). While some of the commentators discussed so far are generally inclined towards the notion that God directly protected the Qur'an, others point out that He employs scholars and reciters, then draw attention to the means of Divine protection. Others expand the textual context of the verse, which is the subject of discussion, and when read very carefully, they imply that Satan cannot interfere when the revelation of the Qur'an comes to the Prophet. Regardless of the form, almost all these commentators associate protection with Divine preservation and do not perceive it only as protection of conveying the revelation from God to the Prophet (vertical), but also include the process that starts with the Companions and will lead to the Day of Judgment (horizontal).[9] I will discuss the subject in the light of Bediuzzaman Said Nursi's (d.1960) explanations with reference to different verses rather than just one (15:9). In this context, my main argument is that this verse and other verses describe the process of Qur'anic revelation without interruption reaching the Prophet. As for the post-Prophetic period, when we look at the efforts of the Muslim intellectual tradition regarding the history of the *Mushaf* and *qiraat*s (variant readings), it is observed that God left the protection of His book to the foresight of the community. I believe in the light of the history of the Qur'anic text that they properly fulfil this mission.

When we look at verses 15:6–8, it is seen that the Prophet was accused of insanity (possessed by jinns) by the polytheists of Mecca due to the claim of revelation and they told him to bring angels to them if he was telling the truth.[10] In verse 15:8, God says 'We do not send the angels down except for a just cause, *bi al-haqq*, and then [the end of] the disbelievers will not be delayed'. The exegetes generally understand the word 'truth' here as *risalat* (revelation) or *adhab* (punishment). Mawardi mentions four possibilities: the Qur'an, prophethood (*risalat*), decreeing one's death and punishment (Mawardi n.d., vol. III, p. 149). All these possibilities have their basis in text and tradition. The first two meanings are preferred. Then verses 15:16–18 explain that jinns and Satan cannot bring revelation from beyond the heavens and these heavenly realms are under the protection of God: 'Indeed, We have placed constellations in the sky, and adorned it for all to see. And We protected it from every accursed Satan (*shaytan rajim*), except the one eavesdropping, who is then pursued by a visible flare (*shihabun mubin*)'. Similar verses are also mentioned in surahs Shu'ara 26:210–212,[11] Saffat 37:6–10,[12] Mulk 67:5[13] and Jinn 71:8–9.[14] While Nursi considers verse 15:9[15] in a different context, he discusses the verses in Surahs Saffat and Mulk within the framework of preservation of the Qur'an. Nursi deals with this subject in the 28th chapter of the *Flashes* (*Lemalar*):

> Spies from among the jinn and Satans eavesdrop on events in the heavens, and like soothsayers, mediums, and some spiritualists, convey news from the World of the Unseen. So that their giving information about the Unseen should not give rise to any doubts when the Qur'an was first revealed, their continual espionage was prevented to a greater extent, and they were repulsed by shooting stars. (Nursi 2007, p. 340)

Nursi's statement "their giving information about the Unseen should not give rise to any doubts when the Qur'an was first revealed" (Nursi 2007, p. 340) is worth dwelling on. Expressions such as *shihaban rasada* (flare lying in wait for them), *shihabun thâqib* (a piercing flare), *harasan shadidan wa shuhuba* (stern guards and shooting stars), *shaytanin marid* (rebellious devil) and *duhuran* (fiercely driven away) mentioned in the verses noted above and in footnotes are clear evidence that the Qur'an was taken under serious protection against external interventions at the beginning of the revelation. I will not dwell on the semantics of these words, which have various nuances in exegetical traditions. However, to better understand Nursi's arguments, I draw attention to the general evaluations about these verses regarding the preservation of the revelation route. For example, Tabari narrates a report about the 8th verse of Surah Jinn. According to this anecdote, the jinns went to Satan and complained about the doors of the heavens that were closed to them and said they were no longer able to listen to the sky. Iblis (Satan) replied, 'The sky is taken under full protection for two reasons: the first is that God will send a sudden punishment to the earth, and the second is that it is the time when God will send a guide and chastening prophet (*nabiyy murshid muslih*)' (Tabari 2000, vol. XXIII, p. 328).

Tabari cites an exegetical report regarding verse 71:9 of the same surah (Jinn) that the heavens are completely protected, and they (the jinns) are barred from any attempt to find out about heavenly news (Tabari 2000, vol. XXIII, p. 328). Imam Maturidi (d.333/944), on the other hand, supports Tabari's interpretation and refers to the verse "Earlier jinns tried to reach heaven [for news], only to find it filled with stern guards and shooting stars" (Maturidi 2005, vol. X, p. 250). Maturidi makes another comment here: "(When the Satans and jinns' attempts to hear from the heaven have completely disappeared), the soothsaying job or task is over" (Maturidi 2005, vol. X, p. 250). This point is important. According to Maturidi, when the door of heaven was closed to them, they could not communicate with their collaborators (soothsayers) on earth; rather, they could not convey the right and wrong information they were supposed to bring (Maturidi 2005, vol. X, p. 250). The fact the oracles could not be fed in terms of celestial knowledge also finished them off. Exegetes indirectly respond to those who call the Prophet a magician or enchanted in this way. Because, with this interpretation, they say the profession in question is no longer possible; therefore, Muhammad is a true prophet not a soothsayer.

Some commentators make the issue more concrete by drawing attention to a well-known narration in different hadith collections:

> According to a hadith, the Prophet Muhammad was sitting once at night with some of his companions when suddenly a meteor shot (shooting star) gave a dazzling light. He asked what the people used to say in the pre-Islamic days when there was such a shot (of meteor). The companions replied that they used to say that very night either a great man had been born or a great man had died. However, the Prophet said that those meteors were shot neither at the death of anyone nor on the birth of anyone. Rather, whenever Almighty God decides to issue a command, His words are transmitted from one group of angels to another throughout the seven heavens. In this process of transmission, the jinn attempt to eavesdrop on what is going on. They snatch what they manage to overhear and carry it to their friends and astrologers). And when the angels see the jinn doing so, they attack them with meteors. (Muslim n.d., vol. IV, p. 1750; Omer 2020), (https://aboutislam.net/multimedia/videos/how-fortune-tellers-and-magicians-work/, accessed on 5 April 2022)

One of the remarkable aspects in the versions of this narration is that the angels carrying the decree of God are described as *hamalat al-arsh*. It is worth emphasising the relationship between the names of these angels, described in Muslim sources from their facial shapes to their numbers, as *hamalat al-arsh* and the naming of the Companions who memorised the Qur'an as *hamalat al-qur'an*. In short, the angels who brought the Divine revelation from heaven to the Prophet and the Companions who transmitted it to the next generation after receiving it from the Prophet; after that, although their existential and ontological dimensions are different, they are all called *hamalat* (carrier).[16] The report describes the descent of Surah An'am that accompanying 70,000 angels should also be evaluated within this framework.[17] Another point that draws attention in exegeses is that *shihab*s (flares) appeared more frequently in Muhammad's prophetic period. This interpretation also shows that jinns and human devils were seriously engaging in attacks and struggles against the process of the revelation of the Qur'an. For this reason, it is possible the problems of the Meccan polytheists regarding the Qur'an or phenomenon of revelation were related to the vertical process of the Prophet receiving the revelation from God via Gabriel rather than the developments after the prophetic transmission of the revelation to the Companions. In any case, I did not come across a noted problem of the polytheists regarding the transmission of the revelation by the Prophet in the sources. In other words, problematising the authentic transmission of the Qur'an is a new phenomenon and occurred in the post-Prophetic period. The main problems in the mind of the Prophet's contemporaries are limited to the route of the revelation from God to the Prophet. In the context of the above-mentioned verses, Nursi discusses why God emphasises this issue so much and what it means 'to stone demons and devils with meteorites'. Of course, in the understanding of *tawhid* in Nursian theology, the heavens and earth are not completely disconnected realms. According to Nursi, the *lawh al-mahfuz* (preserved tablet) is connected to the human heart through the revelation. The material and spiritual realms are also intertwined (involved) and in constant relation with each other. For this reason, the Qur'anic revelation is an important event that connects the earth (*farsh*) to the throne (*arsh*); therefore, the event of the revelation should be processed flawlessly. In this context, Nursi's comment is:

> However, sometimes a minor, particular event occupies a vast world. In whichever corner of the world you listen, you will hear about it. And sometimes some vast mobilization is not against the enemy's forces, but for a show of pomp and majesty. For example, the event of Muhammad (pbuh) and sacred occurrence of the Qur'an's revelation were the most important events in the land of the heavens and were bruited in every corner of it. Then there were more falling stars, which was proclaiming the degree of splendour of the Qur'anic revelation and its glittering sovereignty and the degree of its truthfulness, which could be penetrated by no doubt, and was expressed and illustrated by the sentries posted on the distant, towering bastions of the vast heavens raining down missiles to drive off and repulse the devils. The Qur'an of miraculous exposition expounds and proclaims that cosmic proclamation and alludes to those heavenly signs.

Yes, such a tremendous heavenly sign, and the spying satans, who being made to do battle with the angels although they could have been blown away at the puffing of an angel, was surely to show the majesty of the Qur'anic Revelation's sovereignty. Also, this splendid exposition of the Qur'an and vast heavenly mobilization indicate that there was nowhere the jinns and devils could interfere on the long way from the heart of Muhammad (pbuh) to the world of the heavens and the Sublime Throne, not that the jinns and satans possess some power which drove the inhabitants of the heavens to fight them and defend against them. The Qur'anic Revelation was a truth discussed by all the angels in the heavens; in order, the satans were compelled to rise to the heavens to draw close to it a little but were not successful and were repulsed. This shows that the revelation that came to the heart of Muhammad (pbuh), and Gabriel who came to his presence, and the truths of the Unseen which appeared to his gaze, were sound and straight and could be pierced by no

doubts. The Qur'an of miraculous exposition tells this in miraculous fashion (Nursi 2007, pp. 341–42).

With these quotations, I did not want to draw the wrong conclusion that Nursi understands that only the process of the Qur'anic revelation from God to the Prophet is protected or guaranteed. When read carefully, Nursi implies a prime importance and emphasis on the process of descending the revelation from God to the Prophet than the focus on the post-revelation period at that time. Because it is understood that Meccan polytheists at the time of the Prophet did not have serious concern with the memory of the Arabs and transmission of the Qur'an, which is constantly read in their daily rituals and gatherings, after the revelation.

## 5. Analysis of the Intra-Qur'anic Evidences

Apart from the above-mentioned verses regarding Nursi's inference, other verses support this reasoning and argument. In other words, a wide spectrum of Qur'anic evidence come from the verses about the *lawh mahfuz* (guarded tablet) to the nature and position of Angel Gabriel, and from there to the verses about the status of the heart of the Prophet, the place where the Qur'an was stored for the first time in the earthly domain. For example, the last two verses of Surah Buruj (85:21–22), 'But this is an honoured Qur'an, [Inscribed] in a Preserved Tablet', are worth mentioning in this context. The expression *mahfuz* has two variant readings, *mahfuzin* and *mahfuzun*. In the first one (*fi lawhin mahfuzin*), *maḥfūẓin* is mostly read as a genitive and an attribute of *lawh* (tablet), as recited by nine *qiraat* imams. According to this reading, the meaning is 'in the preserved tablet'. The second one is read *fi lawhin mahfuzun* by Imam Nafi among the ten authentic readings. Qiraat Imam of Madina, Nafi (d.169/785) takes the word (*mahfuz*) as an attribute of the word *qurʾān* in the previous verse. Tabari also records that Ibn Muhaysin (d.123/741), one of the Meccan *qiraat* imams (even though he cannot be among the ten *mutawatir* readers, he is generally considered the eleventh Imam among the 14 *qurra*) also recited it in the form of *mahfuzun*. Noting that both (*mahfuzin* and *mahfuzun*) are common recitations, Tabari explains the recitation of Hijazi *qurra*s as: *bal huwa qur'anun majidun, mahfuzun min al-taghyir, wa al-tabdil fi lawhin* (This is an honoured Qur'an, preserved from change in the (heavenly) tablet) (see Tabari 2000, vol. XXIV, p. 348; Abu Shama 2013, p. 722). These two readings refer to either the existence of the 'preserved/protected Qur'an' in the celestial tablet or the existence of the Qur'an in the 'preserved heavenly tablet'; thus, both variants show the relevant verse emphasises the well-guarded source of the Qur'an. Of course, this does not mean the Qur'an was not preserved when it came out of its original source.

Emphasis on the nature and position of Gabriel is also striking in the Qur'an. For example, verses 16:102,[18] 26:193,[19] 80:15[20] and 81:21[21] point out the messenger (Gabriel) who brought the revelation to the Prophet was immaculate, reliable and holy. These verses contain two significant meanings, one of which is that the vehicle that brings a holy book like the Qur'an to the Prophet is a pure and holy who is free from all kinds of stains and contamination, which indicates the perfection of the messenger (Gabriel) and message (the Qur'an). The second meaning is that the presentation of Gabriel in the form of a clean, reliable spirit is to reveal the uprightness and cleanliness of the Prophet (Yazır 1979, vol. V, p. 443). To put it more clearly, it means that the purest messenger (mediator angel) brings the purest message (the Qur'an) to the purest receiver (Prophet Muhammad).

Just like the nature of Gabriel, it is clearly stated in many verses that the Prophet received and conveyed the revelation flawlessly. First, if we look at the verses about the Prophetic process in receiving the revelation, two verses are worth mentioning: 75:16–17[22] and 87:6.[23] In the first, as stated in the exegesis of *Jalalayn*, it is said that placing the Qur'an on the Prophet's chest (*sadrika*) belongs entirely to God (Mahalli-Suyuti 2010, p. 582). In the second verse, God draws attention to how the Prophet is supposed to preserve the Qur'an (memorise): 'We will make you read it, you will memorize it, so do not tire yourself'. The Prophet was not even allowed to show special will or extra effort for the protection and memorisation of the revelation. This task is assumed by God to place it in the heart of

the Prophet. Another verse supports these meanings, 3:161: 'it is not for any Prophet to deceive (humankind) . . . ' What is more interesting is that exegetes generally make two interpretations regarding the meaning of this verse. According to the first interpretation, the Prophet does not betray by illegally withholding spoils of war. The second interpretation is directly related to our subject: 'the prophet does not betray the revelation' (Bilmen 2019, vol. I, p. 488). As it is seen, the revelation that came perfectly to the Prophet was also perfectly transmitted by him.

Regarding this issue, another verse is even more striking in Surah Baqara (2:97): 'Say, (O Prophet) Whoever is an enemy of Gabriel should know that he revealed this (Qur'an) to your heart by God's Will, confirming what came before it—a guide and good news for the believers.' The late Ottoman and early Turkish Republican commentator Elmalı'lı Muhammed Hamdi Yazır (d.1942) draws attention to the expression *ala qalbika* in the verse and makes a point about the wisdom behind the use of the preposition *ala* (on) instead of *ila* (to) here. He states the preposition *ala* expresses the notion of *isti'la*, which means covering all sides. Thus, the Divine information that comes through the revelation not only sticks to the heart at one point or corner, like the formation of simple feelings such as inspiration (*ilham*) and unexpected meanings (*sunuhat*) that arise spontaneously in the heart. On the contrary, it expresses an irresistible, compulsory scientific reality and Divine wisdom that comes and settles over all kinds of certainty by invalidating all other emotions and perceptions that cover the whole heart. With the usage of the preposition *ala*, it is observed that too much pressure was put on the Prophet's heart and this is his heaviest part. This pressure surrounded his heart from all sides (Yazır 1979, vol. I, p. 360). This point is also mentioned in the Qur'an by the expression *qawlan thaqilan* 'heavy expression' (73:5); therefore, the Prophet, during and after receiving the revelation, used to feel a lot of pain in his head and body. It is reported by various anecdotes that the Prophet frequently had cupping, used henna and took local medicines for this reason. With the preposition *ala*, it is as if God is saying, during the revelation process, He did not leave anything other than the revelation in the Prophet's heart. It is safe to assume the revelation covers every part of the Prophet's cells. We see that the revelation that comes through a solid channel has established its marquee on a solid place in its first stop on earth.

In some verses, it is seen, while God describes the placement of the revelation in the Prophet's heart, He even left out Gabriel in this process. He takes the whole process on Himself and it is seen that He directly connects the heaven and earth. For example, verses 3:58[24] and 27:6[25] are described as if they came directly to the Prophet by God. Especially with the expressions All-Wise and All-Knowing at the end of the verse, it is safe to conclude the scope of communication has expanded tremendously. Although Angel Gabriel (*ruh al-amin*) is believed to bring the words of the Qur'an to the Prophet, the verse points out that God directly informs the subtleties, delicacies, wisdoms, particularities and many secrets to the prophets (Yazır 1979, vol. VI, p. 125). In fact, all these intra-Qur'anic discussions and interpretations show, in the revelation procedure, the bonds between the sender (God), vehicle (Gabriel), message (the Qur'an) and receiver (the Prophet) are very strong, and nothing interferes with this process. On the other hand, some verses imply that God's communication with His Prophet goes beyond the mediator's capacity.

In fact, the whole mode of the revelation case seems to be related to this process. What the Meccan polytheists have difficulty in understanding is the communication between three different ontological beings' levels. It is primarily based on the communication of the Transcendent Being (the Creator) with a human (the Prophet) at a different ontological level, through an intermediary (Gabriel) who is also different from the two former ontological levels and the authenticity of this incoming message. It is seen they do not have any problems with the subsequent processes.[26] From the perspective of the Qur'an, it can easily be said there was no problem in transmitting the revelation to the Companions after they were received by the Prophet. For instance, verses 19:97[27] and 54:17[28] mentioned in Surah Maryam and Qamar are noteworthy examples in this regard. These verses point out that the Qur'an is made easy on the tongue and to remember. The interpretation of

this expression, *wa la-qad yassarna al-qur'ana li al-dhikr*, which occurs four times in Surah al-Qamar, is generally explained as *sahhalnahu li al-hifzi* 'we made it easy for memorisation'. It is also noted the question (*istifham*) in the second part of the verse, *wa hal min muddakir*, is generally explained as a Divine command: 'memorize the Qur'an' (Mahalli-Suyuti 2010, p. 536). In other words, once the part of receiving the revelation by the Prophet, which we consider the difficult part for him, with the help of God is completed and the revelation is placed in his heart, its transmission and memorisation will be much easier. The history of Islam bears witnesses to this with the existence of hundreds of thousands of *huffaz* in every century who know or do not know the language of the Qur'an.

## 6. Conclusive Remarks

After all these analyses of exegetical works, how should one understand the notion of 'Divine protection of the Qur'an'? Considering some of the events that took place in the early period and precautions that were quickly taken, what should our perspective be on the relevant verse? More importantly, how should verse 15:9 be interpreted in the light of the processes that the *Mushaf*s went through in terms of orthography in parallel with the development of writing materials in the following years. The most important thing is how we understand Umar's reaction (the Qur'an will be lost or a large part of it will go) after the martyrs of Yamama (Buti 1999, vol. I, p. 45), although it had not been a long time since the Prophet's death. When we look at evaluations of the commentators, although they frequently brought 'Divine protection' to the agenda, the issue of collection of the Qur'an between two covers after the Prophet and the efforts related to the following developments always remained alive in a corner of their minds. It does not seem consistent, at least when explaining verse 15:9, to bring the falsification of other Divine books as evidence for the Divine protection of the Qur'anic text. As it would be remembered, while explaining the disputes that broke out between the Syrian and Iraqi soldiers of the Muslim army fighting on the borders of Armenia and Azerbaijan to the Caliph Uthman, the companion Abu Huzayfa began his talk by saying, "O caliph! Come to aid of the *ummah* (*adrik*) before they fall. They would have differed in opinion in their book (the Qur'an) similar to those religious communities before them" (Nasa'i 1992, vol. I, p. 67). Abu Huzayfa wanted to see Caliph Uthman take concrete action. Of course, the Qur'an was and is not lost, but it is obvious the disunity and some messiness in the recitations would have damaged the text in the future. For this reason, Abu Huzayfa's request for the Caliph to do something on behalf of the unity of recitation is an issue worth dwelling on. Hajjaj (d.95/714), Umayyad governor of Iraq, had circulated slander among the people that Ibn Zubayr (d.73/692), arch enemy of Umayyad, had falsified the Qur'an during the pilgrimage to make black propaganda about Ibn Zubayr. Ibn Umar (d.73/693), who was on the pilgrimage at that time and heard Hajjaj's words, cried out without hesitation that Hajjaj was lying and remarked that neither ibn Zubayr nor Hajjaj had the power to distort the Qur'an (Dhahabi 1985, vol. III, p. 230). Ibn Umar's confidence in the Qur'anic text that no one could have changed the Qur'an is beyond description. Where does this confidence come from? Is it verse 15:9 or due to the Muslim communities' recording and memorisation of the Qur'an simultaneously from the outset, and their constant reading in daily prayers and other occasions at regular basis? Ibn Umar, who had witnessed both the time of the Prophet Muhammad and the long painstaking period that followed (from the four caliphs to the half of Abd al-Malik's caliphate) is very well aware of this fact. I wonder if the Prophet and his companions, who pointed out linguistic errors even in simple daily conversations and talks, would have allowed wrong readings of the Qur'an.[29]

Unfortunately, some researchers today treat Muslims like the young honeyeaters mentioned at the beginning of this article. Muslims are not young honeyeaters who did not learn or forgot their mating songs. The Divine protection of the Qur'an, as Razi and others have stated, was realised by the permission of God and the efforts of the early Muslims. Existing manuscripts and newly found ones (from San'a to Taskhent, Topkapı to Paris, Tübingen, London, Dublin, St Petersburg) reveal the same truth. Except for scribers'

mistakes, we do not encounter any systematic errors in the Qur'anic manuscripts found in the West and East. This includes the verses on the coins cut in the first century of Islam to thematic verse groups in the Dome of the Rock in Jerusalem. In the first centuries of Islam, Muslims had serious precautious reflexes to protect the Qur'anic text. Every effort should be evaluated within this framework, from the collection of the Qur'an during the reign of Abu Bakr (d.13/634) to the duplication and distribution of the *Mushaf*s by Uthman in an orderly manner, from the efforts of Hajjaj to Ibn Mujahid's (d.324/936) book on variant readings. For instance, we know Caliph Abu Bakr and his assistant Umar did not leave the collection of the Qur'an to chance when they chose Zayd b. Thabit: "Zayd was the one who memorised the Qur'an best, and who knew how to write it best, who understood the abrogating and abrogated verses best, who knew the best recent revelation" (Hamad 2018, p. 313). Umar, the second Caliph, warned Ibn Mas'ud, who read the Qur'an in the Huzayl dialect in Kufa, by saying the Qur'an was revealed in the Quraysh dialect, did not read it according to the Huzayl dialect (Ibn Atiyya 2002, vol. III, p. 243). Caliph Uthman sent Qur'an experts to the central cities along with the copied *Mushaf*s (Tabari 2000, vol. I, p. 60). They do not neglect the theoretical or practical sides of recitation of the Qur'an. Abu Abd al-Rahman al-Sulami (d.73/692?), who went to Kufa, taught the Qur'an there for 40 years (Demirci 1994, vol. X, p. 87). It is known, in the following periods, as a miracle and grace of the Qur'an, the imams of *qiraat* (*qurra*) lived long lives in general and taught the Qur'an for years in the towns where they lived. Imam Nafi, the *qiraat* imam of Madina, had been training students in the Prophet's mosque for 70 years (Altıkulaç 2006, vol. XXXII, p. 287). Abd al-Aziz b Marwan (d. 86/705), father of famous Umayyad Caliph Umar b. Abd al-Aziz (d.101/720), offered large sums of money for any mistakes that could be found in the first period *mushafs*; only one Kufan reciter is mentioned in the sources where he found the word *na'jatan* was written *naj'atan* in verse 38:23 (Asqalani 1998, vol. I, p. 215). It is also worth mentioning, in the early years of Islam, Muslim sources speak of special shops selling *Mushaf*s. Due to the rapid spread of Islam and conversion of many non-Arabs to Islam, we witness the inclusion of auxiliary elements, such as *tashkil*, punctuation (dots) of similar letters and determination of the end of words and verses, to solve the problems encountered in the recitation of the Qur'an. It is also known they wrote them in different colours in the early periods so they would not be confused with the original letters of the *Mushaf*. It is known they developed sciences such as *rasm al-masahif*, *dabt al-masahif* and *khatt al-masahif* under the title of *ilm al-masahif* or *ilm adab kitabat al-masahif* to protect the authenticity and autonomy of the Qur'anic text (Hanash 2018, pp. 619–98). It is seen that numerous works have been written on *al-waqf wa al-ibtida* (pausing and beginning, stops) to determine the *ilm add al-ayah* and stops related to the determination of the number of verses in the *Mushaf*s.[30] When it comes to reciters (*qurra*), narrators (*rawis*) and narrators of narrators (*turuq*), we encounter a perfect system. It has not been possible for any religious text to include significant literature that did not hide or skip any details. This system, which started in the early periods of Islam, dealt not only with the writing of the Qur'an, but also its proper recitation (*tilawat* and *ada*).

In addition, they mention public libraries where only *Mushaf*s were being read by those who did not have money to buy the private *mushaf*. This shows these efforts are not only an intellectual endeavour but also extended to the public to allow them to become familiar with the Qur'anic text. Fuat Sezgin (d.2018), one of the greatest historians of sciences of our century, shows interest in the art of writing among the Muslims of the first century: 'If the Arabic sources about the surprising interest of the early Muslims in the art of writing are examined, the literacy rate of the people living in the first century of Islam (7 CE) in Muslim lands has reached a level that cannot be compared with the Western Middle Ages' (Fuat Sezgin 2008, vol. I, pp. 5–6). The main reason underlying this surprising interest of Muslims towards writing is to serve the Qur'an and its language (Arabic) and to record it in a healthy authentic way. The question that naturally comes to mind here is what is the reason for all these activities when God's promise of protection is obvious? Razi and like-minded exegetes argue that God's action in the world of causes

(*asbab*) is to do this work by means. Every activity that the Muslim community performed in terms of the protections of the Qur'an is actually Divine protection.

So, 'are Muslims going to ignore all efforts with the comfort and slackness of the idea of Divine protection?' There is no doubt the Qur'an has been protected from falsification, alteration and corruption. The Divine statement in the Qur'an is true, but this protection has been realised by the permission of God and ceaseless efforts by the early believers. Today, if Muslims from Northern Kazakhstan (Bulayevo) to West Africa, from the Balkans to the Asia-Pacific read the same text despite their ethnic differences, variety of languages, schools of thought and sects, they owe a lot to the first generations who took care of their scripture. Their understanding of verse 15:9 enabled the followers of the religion of *tawhid* (Oneness of God) to make oneness (*tawhid*) from the word of God (Ghazzali n.d., vol. I, p. 152).

**Funding:** This research received no external funding.

**Acknowledgments:** I am grateful to Peter Riddell for his valuable suggestion and comments on the first draft of this paper.

**Conflicts of Interest:** The author declares no conflict of interest.

## Notes

1.   There are some exceptions written in Arabic in the late 19th and early 20th centuries. *Khulasat al-Bayan fi Ta'lif al-Qur'an* by the Ottoman statesman and intellectual Ahmet Cevdet Pasha (d. 1895) and the great scholar from Kazan (Russia) Musa Jar Allah Bigiev's (d. 1949) *Tarikh al-Qur'an wa al-Masahif* are two of them. The *Maqalat* of Muhammad Zahid al-Kawthari may also be mentioned. It contains some articles on the history of the Qur'an. Kawthari, *Maqalat* (first publications: Cairo 1952). This book consists of various articles of the author compiled by his students. There are also few Iranian scholars who have written about the history of the Qur'an.

2.   It is the fifteenth surah in the *Mushaf* and the fifty-fourth in the revelation order. It was revealed during the Makkan period after chapter Joseph and before chapter An'am.

3.   5:44: ' . . . So too did the rabbis and scholars judge according to Allah's Book, with which they were entrusted and of which they were made keepers . . . '

4.   5:67: ' . . . Allah will [certainly] protect you (Muhammad) from the people . . . '

5.   4:82: ' . . . Had it been from anyone other than Allah, they would have certainly found in it many inconsistencies.'

6.   The question of why a bookstore and not a mosque may come to mind. Probably, our Andalusian exegete consider this Jewish man religiously unclean and did not want to let him into the mosque.

7.   Professor Peter Riddell noted that Khazin is hardly original and does not add much to Baghawi, he argued that Khazin is really an adapted version of Baghawi (See his forthcoming chapter in *Handbook of Qur'ānic Hermeneutics* by Prof. Georges Tamer (edited in collaboration with the publisher Walter de Gruyter)) (Riddell, forthcoming).

8.   *Bahru al-Ulum* is the Qur'anic exegesis of the famous commentator Ala al-Din Ali b Yahyâ al-Samarqandî (d. 860/1456), which was widely read by the lay people in the Ottoman lands.

9.   It is interesting that Qur'anic exegetes did not consider the timing of this verse's revelation (15:9). This verse was revealed before most of the other Qur'anic chapters. For this reason, the verse talks about the polytheists' hesitancy and suspicion about the coming of revelations rather than the *mushaf* that had not yet been brought between two covers.

10.   15:6–7: 'They say, "O you to whom the Reminder is revealed! You must be insane! Why do you not bring us the angels, if what you say is true?"'

11.   26:210–12: 'It was not the devils who brought this [Quran] down: it is not for them [to do so], nor can they, for they are strictly barred from [even] overhearing [it]'.

12.   37:6–10: 'Indeed, We have adorned the lowest heaven with the stars for decoration and [for] protection from every rebellious devil. They cannot listen to the highest assembly [of angels] for they are pelted from every side, [fiercely] driven away. And they will suffer an everlasting torment. But whoever manages to stealthily eavesdrop is [instantly] pursued by a piercing flare'.

13.   67:5: 'And indeed, We adorned the lowest heaven with [stars like] lamps, and made them [as missiles] for stoning [eavesdropping] devils, for whom We have also prepared the torment of the Blaze'.

14.   71:8–9: '[Earlier] we tried to reach heaven [for news], only to find it filled with stern guards and shooting stars. We used to take up positions there for eavesdropping, but whoever dares eavesdrop now will find a flare lying in wait for them'.

15.   Nursi talks about Qur'anic protection rather than preservation of the Qur'an in the context of 15:9. First he discusses the notion of fear in human beings and says 'One of the strongest and most basic emotions in human is the sense of fear. Scheming

oppressors profit greatly from the vein of fear . . . My brothers! If those who today to the atheists attack you by frightening you into giving up your sacred *jihad* of the word, say to them: "We are the party of the Qur'an." According to the verse, "We have, without doubt, sent down the Message; and We will assuredly guard it," (15:9) we are in the citadel of the Qur'an' (Nursi 1997, p. 437). For Nursi, the Qur'an is a protector; it protects believers from every kind of worldly worry. When it comes to preservation of the Qur'an, it is *res judicata* for Nursi. (Some of the translation of Nursi's texts is taken from online English version. See www.erisale.com/index.jsp?locale=en, accessed on 7 April 2022).

16  Mystically speaking, there are two *arsh*s and both are very closely related to each other. One is the heavenly *arsh*, which is unique to God, and the other is earthly *arsh* where God is hosted in earthly realm and the Qur'an is preserved, namely human heart.

17  Regarding this surah, the Messenger of God said, 'The chapter of *al-An'am* was revealed to me with a group of 70,000 angels accompanied its descent with *tasbih* and *hamd* (the words of glorification and praise) of God' (Ibn Kathir 1998, *Tafsir al-Qur'an al-Azim*, vol. III, p. 238). Ibn Abbas said: 'Surah An'am was revealed in Mecca all at once. Seventy thousand angels accompanied this Surah. While the Surah descended, the angels filled the gap between the two mountains of Mecca. The Messenger of Allah (pbuh) called the scribers, and they wrote down the whole Surah that night, except for the 6 verses that were revealed in Madinah.'

18  'Say: The holy Spirit hath revealed it from thy Lord with truth, that it may confirm (the faith of) those who believe, and as guidance and good tidings for those who have surrendered (to Allah) (*ruh al-quds*)'.

19  'Which the True Spirit hath brought down (*al-ruh al-amin*)'.

20  '(Set down) by scribes (*bi-aydi safarah*)'.

21  'One to be obeyed, and trustworthy (*muta'in thamma amin*)'.

22  'Do not rush your tongue trying to memorize (a revelation of) the Quran. It is certainly upon Us to (make you) memorize and recite it'.

23  'We will have you recite (the Quran, O Prophet), so you will not forget (any of it)'.

24  'We recite (all) this to you (O Prophet) as one of the signs and (as) a wise reminder' *dhalika natluhu alayka* (We recite to you).

25  'And indeed, you (O Prophet) are receiving the Quran from the One (Who is) All-Wise, All-Knowing'.

26  If the Qur'an is to be followed, the Jews of Madina had objections, as stated in verse 6:91: 'And they have not shown God proper reverence when they said, "God has revealed nothing to any human being." Say, (Prophet), "Who then revealed the Book brought forth by Moses as a light and guidance for people, which you split into separate sheets—revealing some and hiding much? You have been taught (through this Quran) what neither you nor your forefathers knew." Say, (O Prophet) "God (revealed it)!" Then leave them to amuse themselves with falsehood.' Just like the previous prophets, it is clearly stated in the verse that Prophet Muhammad received revelations and these were written during his lifetime. This is something known. Mentioned after a series of verses in Surah Isra (17:93), ' . . . we will not believe in your ascension until you bring down to us a book that we can read. Say, "Glory be to my Lord! Am I not only a human messenger?"' also shows this. In fact, verse 74:52 also refers to this expectation: 'each one of them wishes to be given a (personal) letter (from God) for all (to read)'. This verse in Surah Muddaththir is noteworthy in that it points to the perception of the book in people's minds.

27  'Indeed, We have made this (Quran) easy in your own language (O Prophet) so with it you may give good news to the righteous and warn those who are contentious'.

28  'And We have certainly made the Quran easy to remember. So is there anyone who will be mindful?'

29  Umm Ayman addressed the Companions who suffered a partial defeat on the day of Hunayn *sabbata Allahu aqdamakum* (May Allah cut your feet or give you rest) instead of saying *thabbata Allahu aqdamakum*, which means 'May Allah make your feet stable'. When the Prophet heard Umm Ayman's mispronunciation, he said to her 'O Umm Ayman, please be quiet' (Ibn Sa'd 1990, *al-Tabaqat al-Kubra*, vol. VIII, p. 180). Umm Ayman was an Abyssinian woman so she could not speak Arabic very well.

30  Ibn Nadim talks about books composed about the identification of the number of verses of the Qur'an and Qur'anic stoppings, *al-kutub al-mu'allafa fi adad ay al-qur'an, al-kutub al-mu'allafa fi al-waqf wa al-ibtida* (Ibn Nadim 1997, al-Fihrist, vol. I, pp. 55–57).

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
