# Peer review of "Revisiting the Meaning of the Divine Preservation of the Qur’an: With Special References to Verse 15:9"

_religions, doi:10.3390/rel13111064_

Round 1

Reviewer 1 Report

Specific observations:

1.     The article begins with anecdotal and superficial descriptions about Western scholarship on the Qur’an, especially concerning the history of the Qur’anic text. It raises some concerns pertaining to “a lot of noises” caused by the critical scholarship. The problem with this superficial reference to Western scholarship is that it does not present a full picture of the dynamic of scholarly conversations on the Muslim scripture. I don’t think that the author’s description of the Western critical approach is helpful for the general Muslim audience, let alone for specialist in the field. The article does a disservice to the rich discussions which make Qur’anic studies a vibrant field of research.

2.     The article strives to discuss exegetical works from the classical to modern periods chronologically, from Muqatil b. Sulayman to Sayyid Qutb. However, it ignores major works such as Ibn Kathir or Suyuti (al-Durr al-manthur). When it comes to the modern period, the article only elucidates Qutb’s work.

3.     It is worth noting that the author spends great of time talking about Said Nursi’s tafsir in a separate section. I totally understand that such selection is a subjective endeavor; however, there should be some reasons why he is treated differently from others, like Qutb or medieval commentators such as Tabari, Zamakhsshari, and others. The article does not say anything about the significance of Nursi’s exegetical work compared with many other exegetes whose works are discussed in the article.

4.     Concerning the selection of tafsir, the article seems to choose a tafsir that represents a certain period of time, which does make sense as the author claims to discuss Q 15:9 through the lens of exegetes in successive generations. However, the selection is confusing not only because the author does not offer any reasoning for his/her selection (why focusing on tafsir in Arabic alone or why focusing on Sunni tafsir and disregards Shi’i, for instance), but also because there is a huge gap from the 16th century to the 20thcentury in which no exegetical works were discussed at all. There are many great tafsirs during this period which deserve attention, such as the works of Shawkani, Abu al-Thana’ al-Alusi, Abduh/Rashid Rida, etc. 

5.     There are many interesting ideas that are not fully developed. For instance, in the abstract, the author says “The use of the same Qur’anic text by Sunnis, Shi’ites, Ibadis, etc., from northern Kazakhstan to West Africa, from Asia Pacific to the Balkans, is the clearest proof that Muslims are preserving entrusted to them.” Well, this argument is often presented by conservative Muslims over and again. This idea is not developed at all in the article! The fact that all Muslims of various regions and of different theological divide use the same Qur’an today does not tell anything about its preservation from the time of the Prophet to later generations. The argument can only be relevant for the time when the Qur’an was put into a mushaf (in writing), but not before then. From a historical critical perspective, we simply do not know what happened before the Qur’an was codified in a mushaf. Moreover, the Muslims use the same Qur’an with which one? The standard/existing Qur’an can be traced back to the 1924 Cairo edition.

6.     The author entertains the question of i’jaz (the Quran as miracle) which prevents it from being falsified (with additions or removal). However, there is a wholesale discussion among Muslim exegetes on this question in the context of the Qur’an’s transmission. Is the Qur’an a preserved text because it is miraculous or because it is transmitted through tawatur? What is the parameter for its preservation? This issue seems to slip the author’s radar.

7.   Also, controversies over basmalah (p. 6) deserve a fuller treatment than it is presented in the article. The Andalusian exegete al-Qurtubi discusses this question in several pages in his tafsir

Author Response

First of all, I thank the Reviewer for this valuable evaluations:

  1. The article begins with anecdotal and superficial descriptions about Western scholarship on the Qur’an, especially concerning the history of the Qur’anic text. It raises some concerns pertaining to “a lot of noises” caused by the critical scholarship. The problem with this superficial reference to Western scholarship is that it does not present a full picture of the dynamic of scholarly conversations on the Muslim scripture. I don’t think that the author’s description of the Western critical approach is helpful for the general Muslim audience, let alone for specialist in the field. The article does a disservice to the rich discussions which make Qur’anic studies a vibrant field of research.

-I removed the phrase “a lot of noises” from the text of the article. As someone who did his PhD in the West and has benefited greatly from Western scholarship, I do not criticize Western Qur'an studies with a wholesale approach. In my opinion, I observed that some revisionist academics prioritize the methodology they constructed rather than the text of the Qur'an. I do not want to go into detail regarding this subject. I just stated in the introduction that I do not find the approaches of both Muslims and some Western writers very healthy. There are also many Western scholars who criticise the revisionist way of engaging with the Qur’anic text and history.

  1. The article strives to discuss exegetical works from the classical to modern periods chronologically, from Muqatil b. Sulayman to Sayyid Qutb. However, it ignores major works such as Ibn Kathir or Suyuti (al-Durr al-manthur). When it comes to the modern period, the article only elucidates Qutb’s work.

- Ibn Kathir's evaluation about the verse is already mentioned in the article. Please see p.3. As for Suyuti's Durr al-Manthur, I did not quote. Unfortunately, Suyuti cites a very short exegetical report on the relevant verse. That's why I couldn't get to the article.

  1. It is worth noting that the author spends great of time talking about Said Nursi’s tafsir in a separate section. I totally understand that such selection is a subjective endeavor; however, there should be some reasons why he is treated differently from others, like Qutb or medieval commentators such as Tabari, Zamakhsshari, and others. The article does not say anything about the significance of Nursi’s exegetical work compared with many other exegetes whose works are discussed in the article

-Upon this advice, I changed the structure and evaluated Said Nursi not under a separate sub-title, but under the modern period sub-title (together with Qutub and Tabatabaei). The arguments I tried to discuss in the article on the vertical and horizontal preservation of the revelation are inspired by some of Nursi's assessments and my re-reading of the relevant verses.

  1. Concerning the selection of tafsir, the article seems to choose a tafsir that represents a certain period of time, which does make sense as the author claims to discuss Q 15:9 through the lens of exegetes in successive generations. However, the selection is confusing not only because the author does not offer any reasoning for his/her selection (why focusing on tafsir in Arabic alone or why focusing on Sunni tafsir and disregards Shi’i, for instance), but also because there is a huge gap from the 16th century to the 20thcentury in which no exegetical works were discussed at all. There are many great tafsirs during this period which deserve attention, such as the works of Shawkani, Abu al-Thana’ al-Alusi, Abduh/Rashid Rida, etc.

-I added the evaluations of the Shiite commentators such as Tusi, Tabarsi and Tabatabaei. You can see it in the text.  I had already covered Alusi extensively. As for Shawkani, his comments on the verse are very short and does not say much. Unfortunately, Tafsir al-Manar by Abduh and Rida does not contain the relevant verse. Please see the pages 3 and 10-12 for Shi’ite commentators and for Alusi see pp. 8-9.

  1. There are many interesting ideas that are not fully developed. For instance, in the abstract, the author says “The use of the same Qur’anic text by Sunnis, Shi’ites, Ibadis, etc., from northern Kazakhstan to West Africa, from Asia Pacific to the Balkans, is the clearest proof that Muslims are preserving entrusted to them.” Well, this argument is often presented by conservative Muslims over and again. This idea is not developed at all in the article! The fact that all Muslims of various regions and of different theological divide use the same Qur’an today does not tell anything about its preservation from the time of the Prophet to later generations. The argument can only be relevant for the time when the Qur’an was put into a mushaf (in writing), but not before then. From a historical critical perspective, we simply do not know what happened before the Qur’an was codified in a mushaf. Moreover, the Muslims use the same Qur’an with which one? The standard/existing Qur’an can be traced back to the 1924 Cairo edition.

-The article is quite long, there are many issues that need to be expanded, I agree with some of Reviewer’s evaluations about this article, at least we do not have much information about the pre-Uthmanic text of the Qur’an, but I could not understand of the reviewer’s linking the standard version of the Qur'an with 1924. Apart from the differences in variant readings and unsystematic scriber’s errors, were Muslims reading another Qur'anic text before 1924?  

  1. The author entertains the question of i’jaz (the Quran as miracle) which prevents it from being falsified (with additions or removal). However, there is a wholesale discussion among Muslim exegetes on this question in the context of the Qur’an’s transmission. Is the Qur’an a preserved text because it is miraculous or because it is transmitted through tawatur? What is the parameter for its preservation? This issue seems to slip the author’s radar

-Reviewer is absolutely right in this regard. This is a very huge topic. Honestly, after reading Tabatabaei, I understood the problem better. For Tabatabaei at least, i'jaz is more important than tawatur and ijma. However, it seems the Sunni tradition cares about both. It is a subject that needs to be studied more broadly to see the nuances. I thank sincerely the Reviewer for drawing my attention to this nuanced aspect.

  1. Also, controversies over basmalah (p. 6) deserve a fuller treatment than it is presented in the article. The Andalusian exegete al-Qurtubi discusses this question in several pages in his tafsir.

-Again, another important point which needs to be studied in detail but now it is beyond the limitation of this article.

Reviewer 2 Report

This research addresses a very important theme in Qur'anic studies. The abstract announces the idea of an analysis of the interpretations of verse 15:9 from linguistic, historical, theological and sociological points of view through various eras. This orientation could have been carried out in a more chronological way, the article would gain more readability. The author's effort in this work is praiseworthy, nevertheless, we have the impression that sometimes the periods overlap, giving rise to a kind of reading that requires great agility on the part of the reader to follow the interpretations of the exegetes called upon in the article. I take as an example the 2nd part, which is in fact the 1st after the introduction. From my point of view, it is too dense; it is necessary to specify from the outset what the author is trying to demonstrate through the presentation of the various exegeses.
In this regard, I propose that the introduction and the conclusion be left unnumbered, and that only the parts properly speaking be numbered.
The title of the third part, "The understanding of the verse during post-Razi period" is too broad, it is about the exegesis in the classical age. Sayyid Qutub's reading is part of the modern age, even if it is different from Said Nursi's assessment.  This division between the classical age and the modern age seems to me more coherent.
All these readings are prompted by issues that it would be interesting for the author to clarify. From one reading to another, the political and religious issues change and sometimes contradict each other. How does the same verse yield different interpretations from one another?
In general, however, this research work deserves to be published, as it constitutes a step forward in the reading of one of the key verses of the Qur'an that relates to its preservation and protection.

Author Response

 1. it is necessary to specify from the outset what the author is trying to demonstrate through the presentation of the various exegeses.
In this regard, I propose that the introduction and the conclusion be left unnumbered, and that only the parts properly speaking be numbered.
DONE

 2. we have the impression that sometimes the periods overlap

In general, I tried to give the explanations of the commentators chronologically, but rarely I went outside the chronological framework when there were the same approaches. For example, citing Ibn Kathir before Razi.

3. The title of the third part, "The understanding of the verse during post-Razi period" is too broad, it is about the exegesis in the classical age. Sayyid Qutub's reading is part of the modern age, even if it is different from Said Nursi's assessment.  This division between the classical age and the modern age seems to me more coherent.
DONE

Reviewer 3 Report

1. I recommend staying away from making "value judgements" such as: interesting point, more appropriate interpretation, interesting approach, so and do handled the subject best, etc... 

2. Under the Nursi heading, you present his argument among others. I suggest changing the heading to more general heading to encompass all the points and not just Nursi's opinion. 

3. I did not see any reference made to the Shi'ite sources. In this kind of research it is critical that both Sunnite and Shi'ite exegetical works are examined. I would suggest, at the very minimum, to include Tabatabei's opinion on this as he is the most celebrated Shi'ite Qur'anic commentator of the modern era. 

4. There was a mention of 'temple' when referred to Christians, this should be changed to church. 

Author Response

  1. I recommend staying away from making "value judgements" such as: interesting point, more appropriate interpretation, interesting approach, so and do handled the subject best, etc... 

Thanks for drawing my attention to this issue. I deleted 8-10 of them.

  1. Under the Nursi heading, you present his argument among others. I suggest changing the heading to more general heading to encompass all the points and not just Nursi's opinion. 

DONE

  1. I did not see any reference made to the Shi'ite sources. In this kind of research it is critical that both Sunnite and Shi'ite exegetical works are examined. I would suggest, at the very minimum, to include Tabatabei's opinion on this as he is the most celebrated Shi'ite Qur'anic commentator of the modern era. 

I have included arguments by Tusi, Tabarsi and Tabatabaei on the subject. Tusi and Tabarsi's exegesis are rather short, but Tabatabaei's is very broad. Thanks again for pointing out Tabatabaei. Please see the pages 3 and 10-12.

  1. There was a mention of 'temple' when referred to Christians, this should be changed to church. 

DONE (Unfortunately, Qurtubi calls the Jewish worship place al-kanisa, and the Christian worship place (al-biya').